# Selection and Characterization of Monoclonal Antibodies Targeting Middle East Respiratory Syndrome Coronavirus through a Human Synthetic Fab Phage Display Library Panning

**DOI:** 10.3390/antib8030042

**Published:** 2019-07-31

**Authors:** Yoonji Kim, Hansaem Lee, Keunwan Park, Sora Park, Ju-Hyeon Lim, Min Kyung So, Hye-Min Woo, Hyemin Ko, Jeong-Min Lee, Sun Hee Lim, Byoung Joon Ko, Yeon-Su Park, So-Young Choi, Du Hyun Song, Joo-Yeon Lee, Sung Soon Kim, Dae Young Kim

**Affiliations:** 1New Drug Development Center, Osong Medical Innovation Foundation, Cheongju-si, Chungcheongbuk-do 28160, Korea; 2Korea Center for Disease Control, Osong Health Technology Administration Complex, Cheongju-si, Chungcheongbuk-do 28159, Korea; 3Plexense, Inc., Yongin-si, Gyeonggi-do 441-813, Korea

**Keywords:** MERS-CoV, spike protein, S2 subunit, phage display, monoclonal antibody

## Abstract

Since its first report in the Middle East in 2012, the Middle East respiratory syndrome-coronavirus (MERS-CoV) has become a global concern due to the high morbidity and mortality of individuals infected with the virus. Although the majority of MERS-CoV cases have been reported in Saudi Arabia, the overall risk in areas outside the Middle East remains significant as inside Saudi Arabia. Additional pandemics of MERS-CoV are expected, and thus novel tools and reagents for therapy and diagnosis are urgently needed. Here, we used phage display to develop novel monoclonal antibodies (mAbs) that target MERS-CoV. A human Fab phage display library was panned against the S2 subunit of the MERS-CoV spike protein (MERS-S2P), yielding three unique Fabs (S2A3, S2A6, and S2D5). The Fabs had moderate apparent affinities (Half maximal effective concentration (*EC*_50_ = 123–421 nM) for MERS-S2P, showed no cross-reactivity to spike proteins from other CoVs, and were non-aggregating and thermostable (*T*_m_ = 61.5–80.4 °C). Reformatting the Fabs into IgGs (Immunoglobulin Gs) greatly increased their apparent affinities (*K*_D_ = 0.17–1.2 nM), presumably due to the effects of avidity. These apparent affinities were notably higher than that of a previously reported anti-MERS-CoV S2 reference mAb (*K*_D_ = 8.7 nM). Furthermore, two of the three mAbs (S2A3 and S2D5) bound only MERS-CoV (Erasmus Medical Center (EMC)) and not other CoVs, reflecting their high binding specificity. However, the mAbs lacked MERS-CoV neutralizing activity. Given their high affinity, specificity, and desirable stabilities, we anticipate that these anti-MERS-CoV mAbs would be suitable reagents for developing antibody-based diagnostics in laboratory or hospital settings for point-of-care testing.

## 1. Introduction

Since its first reported isolation from a patient in Saudi Arabia in 2012, the Middle East respiratory syndrome coronavirus (MERS-CoV) has posed a global threat to public health. A pandemic outbreak was initiated in South Korea in 2015 and outbreaks still sporadically occur in various regions, including the Middle East [1,2]. MERS-CoV causes severe respiratory symptoms accompanied by high fever, cough, and severe pneumonia, and mortality rates can be as high as 35% [3]. The overall risk in areas outside Saudi Arabia, the country in which most of the cases have been reported, remains as significant as inside Saudi Arabia, and thus novel tools and reagents for therapy and diagnosis are urgently needed.

Among the four coronavirus genera (alpha, beta, gamma, and delta), MERS-CoV, along with severe acute respiratory syndrome coronavirus (SARS-CoV), belongs to the betacoronaviruses. MERS-CoV is an enveloped RNA virus with a positive-sense, single-stranded RNA genome that encodes structural proteins including the spike (S) protein and the nucleocapsid protein (NP) [4]. Like all coronaviruses, the S protein, present on the surface of the MERS-CoV virion, plays a critical role in the viral entry into host cells [5,6]. The S protein consists of two non-covalently associated subunits, S1 and S2, and trimerizes to form a prefusion spike [7,8,9]. Upon entry of the virus into cells, the receptor-binding domain (RBD) of the S1 subunit recognizes the host cell receptors dipeptidyl peptidase 4 (DPP4) or CD26 [10,11,12], while the S2 subunit has a role in viral fusion with host cell membranes through the interplay of two heptad repeats (HR1 and HR2) and a transmembrane domain on the S2 subunit [6,13].

Vaccines have been the most effective prophylactic tools against viral infections, although their long development time and inability to confer immediate protection or therapy are significant drawbacks [14]. Neutralizing monoclonal antibodies (mAbs) are rapidly emerging as an alternative approach complementing vaccines against viral infections [14]. Due to its critical role in the interaction with host cell receptors, tremendous efforts have been focused on the discovery of neutralizing mAbs against the RBD of the MERS-CoV S1 subunit [15,16,17,18]. Strategies have included phage or yeast display library selection; immunization of animals, such as humanized mice and dromedary camels; and antibody isolation from human survivors of MERS-CoV infection. However, a drawback of the mAbs is their costly and time-consuming production in mammalian cell lines. Moreover, viruses can easily evolve to generate RBD variants with mutations that avoid immune responses [19,20]. However, such mutations are less common in the S2 subunit, which makes it an attractive target for therapeutic purposes. Indeed, the first MERS-CoV S2 subunit (MERS-S2P)-targeting antibody, called G4, was generated using mouse immunization and was demonstrated to recognize a variable loop in the S2 connector domain and to neutralize infection by the virus [13,21].

Although anti-MERS-CoV therapies are urgently needed, there is currently no anti-viral drug or vaccine approved for human use. Therefore, the importance of developing reliable diagnostic tests is rapidly becoming clear to manage and control a viral spread. Indeed, various diagnostic tests have been developed since the first description of the virus. Real-time PCR (RT-PCR), accompanied with viral RNA sequencing, is the primary method used to detect viral infection, although there are obvious limitations of this test such as a long turnaround time [1,22,23,24,25,26,27]. Serological tests relying on antibodies specific to viral antigens, such as the NP, can also be used by either capturing viral antigens through enzyme-linked immunosorbent assay (ELISA) and immunochromatographic tests (ICTs) [28,29,30] or by detecting anti-MERS-CoV antibodies through ELISA, indirect fluorescence antibody assay (IFA), Western blotting, and neutralization assays [31,32,33,34,35]. 

In this study, we panned a synthetic human Fab phage display library against MERS-S2P and obtained monoclonal antibodies (mAbs) that specifically detected the viral S2 subunit. We also described the use of these mAbs for the development of a novel ELISA format called ACCEL ELISA™, which we anticipate can be used as a rapid diagnostic test with a high sensitivity for point-of-care testing (POCT) of MERS-CoV infection.

## 2. Materials and Methods

### 2.1. Library Panning, Phage ELISA, and Production of Fabs 

An in-house produced human synthetic Fab phage display library (KFab-I), which was built on a human VH3 and Vk1 germline-based scaffold with randomized complementarity-determining regions (CDRs) (unpublished data), was used for the selection of specific binders against the S2 subunit of MERS-CoV spike protein (MERS-S2P, provided by Dr. Lee from KCDC (Korea Center for Disease Control, Cheongju-si, Chungcheongbuk-do, Republic of Korea)). Briefly describing the MERS-S2P production, MERS-S2P (amino acids 752-1288 amino acids), tagged with C-terminal 6× His, was expressed in insect Sf9 cells by using a recombinant baculovirus (AB vector, San Diego, CA, USA) and was purified using HisTrap HP Ni-NTA affinity chromatography (GE Healthcare, Uppsala, Sweden) and HiLoad 26/60 Superdex 200 size exclusion chromatography (GE Healthcare). Briefly, to perform panning, t MERS-S2P was immobilized in immunotubes (Nunc, Rochester, NY, USA) at a concentration of 10 μg/mL in PBS (phosphate-buffered saline, pH 7.4) at 4 °C for 18 h. After rinsing twice with tap water, the immunotubes were blocked with 5% skim milk in PBS for 1 h at room temperature (RT). At the same time, the phage library was incubated in 2% skim milk in PBS for 1 h at RT. Blocked phages were transferred to the immunotubes coated with MERS-S2P and incubated for 1 h at 37 °C before washing three times with PBS-T (PBS containing 0.05% Tween 20). Bound phages were eluted from immunotubes with 100 mM triethylamine for 10 min at RT, followed by neutralization with 1 M Tris-HCl (pH 7.4). Neutralized eluted phages were transferred to mid-log-phase *Escherichia coli* (*E. coli*) TG1 cells and incubated for 1 h at 37 °C with gentle rotation (120 rpm). The infected TG1 cells were spread on 2× YT agar plates supplemented with 2% glucose and 100 μg/mL ampicillin, and incubated overnight at 37 °C. The colonies were collected via scraping with 6 mL of 2× YT medium. A total of 50 mL of 2× YT supplemented with 2% glucose and 100 μg/mL ampicillin were inoculated with the scraped cells to yield an OD_600_, the optical density at 600 nm, of 0.05 to 0.1 and incubated at 37 °C with shaking (220 rpm) until the OD_600_ reached 0.5. Then, the culture was infected with VCSM13 helper phage, a derivative of M13KO7 helper phage (provided by Dr. Hong from Kangwon National University, Chuncheon-si, Gangwon-do, Republic of Korea) at a multiplicity of infection of 20:1. After incubation for 1 h at 37 °C with gentle rotation (120 rpm), kanamycin was added to a final concentration of 70 μg/mL and the culture was grown overnight at 30 °C with shaking (220 rpm). Cells were then centrifuged at 24,793× g for 30 min and the supernatant was passed through a 0.22 μm filter. Phage particles were precipitated using one-fifth of the volume of the precipitation buffer (20% Polyethylene glycol 8000 (PEG 8000) (Sigma-Aldrich, St. Louis, MO, USA), 15% NaCl) for 30 min on ice. The precipitated phages were pelleted using centrifugation at 24,793× g for 30 min and resuspended in PBS. The phage suspension was used for the next round of panning. For stringent selections, the number of washing steps was gradually increased and the amount of antigen for immobilization was decreased: first round—10 μg/mL, second round—5 μg/mL, and third round—2.5 μg/mL in PBS (1 mL).

### 2.2. Monoclonal Phage ELISA 

Monoclonal phage ELISA was performed after three rounds of panning. Several 96-Well Half-Area Microplates (Corning, New York, NY, USA) were coated overnight at 4 °C with 30 μL per well of 1 μg/mL MERS-S2P and each well was blocked with 5% skim milk in PBS for 1 h at RT. The amplified phages of 94 individual clones from the third round of panning were added and incubated for 1 h at 37 °C. After washing four times with PBS-T, horseradish peroxidase (HRP)-conjugated anti-M13 antibody (1:5000, Sino Biological, Beijing, China) was incubated for 1 h at 37 °C. After washing four times with PBS-T, 3,3’,5,5’-Tetramethylbenzidine (TMB) substrate solution (Sigma-Aldrich) was added for 8 min, and the reaction was stopped with 1 N sulfuric acid (Merck, Darmstadt, Germany). Absorbance was measured at 450 nm using a SpectraMax 190 Microplate Reader (Molecular Devices, Sunnydale, CA, USA). To demonstrate MERS-S2P specificity, monoclonal phage ELISA was performed with a SARS-CoV spike protein (Abcam, Cambridge, U.K.) and human coronavirus HKU1 spike protein (Sino Biological) as controls.

### 2.3. Production of Fab Proteins 

The in-house bacterial expression vector (pKFAB) was used to construct Fab expression vectors that encode VH-CH1 tagged with a C-terminal Strep-tag. Fab fragments genes were amplified using a polymerase chain reaction (PCR). The PCR products were purified using a QIAquick PCR Purification Kit (QIAGEN, Hilden, Germany) and digested with *Sfi*I (New England Biolabs, Ipswich, MA, USA). The digestion products were separated on a 1% agarose gel and the single band was purified with a QIAquick Gel Extraction Kit (QIAGEN, Hilden, Germany). The fragments were ligated into similarly-digested vector fragments using T4 DNA ligase (New England Biolabs), followed by transforming *E. coli* DH5α competent cells (Enzynomics, Daejeon, Korea) were transformed with the ligation mixtures. Individual colonies of the transformed cells were isolated and the sequences of isolated clones were verified. Top10F’ Competent Cells (Invitrogen, Carlsbad, CA, USA) were transformed with the Fab expression vectors and the transformants were grown in 200 mL of Terrific Broth (TB) media supplemented with 100 μg/mL ampicillin at 37 °C with shaking (220 rpm) until the OD_600_ reached 0.5. The log-phase cultures were then induced with 0.5 mM isopropyl β-D-1-thiogalactopyranoside (IPTG) and incubated overnight at 30 °C with shaking (220 rpm). The cells were collected and resuspended in 16 mL of 1× TES (50 mM Tris-HCl, 1 mM ethylenediaminetetraacetic acid (EDTA), 20% sucrose, pH 8.0). After incubation for 30 min on ice, 24 mL of 0.2× TES was added and incubated for 1 h on ice. The periplasmic fractions were collected after centrifugation at 24,793× g for 30 min and filtered through a 0.22 μm filter. The periplasmic extracts were loaded on a column packed with 0.5 mL of Strep-Tactin XT (IBA Lifesciences, Goettingen, Germany). The column was washed with 10 column volumes (CVs) of Buffer W (IBA Lifesciences) and eluted with 5 CVs of Buffer BXT (IBA Lifesciences). The eluted proteins were concentrated and buffer-exchanged with PBS using Amicon Ultra-15 Centrifuge Filter Units (Millipore, Carrigtwohill, Co., Cork, Ireland).

### 2.4. Determination of Apparent Affinity Using ELISA

Several 96-Well Half-Area Microplates were coated overnight at 4 °C with 30 μL per well of 2 μg/mL MERS-S2P. After rinsing twice with tap water, the wells were blocked with 5% skim milk in PBS for 1 h at RT. Serially diluted anti-MERS-S2P Sterp-tagged Fabs were added and incubated for 1 h at 37 °C. After washing four times with PBS-T, HRP-conjugated StrepMAB-Classic (1:10,000, IBA Lifesciences) was added to the plates and incubated for 1 h at 37 °C. After washing four times with PBS-T, TMB substrate solution was incubated for 8 min, and the reaction was stopped with 1 N sulfuric acid. Absorbance was measured at 450 nm using a SpectraMax 190 Microplate Reader. A plot was created using a four-parameter curve fit algorithm and half maximal effective concentration (*EC*_50_) values were determined accordingly.

### 2.5. Determination of Melting Temperature Using a Protein Thermal Shift (PTS) Assay 

To each well of a MicroAmp Fast Optical 96-Well Reaction Plate (Applied Biosystems, Foster City, CA, USA), 18 μL of anti-MERS-S2P Fab and 2 μL of Protein Thermal Shift Dye (10×, Applied Biosystems, Foster City, CA, USA) were added. As a negative control, PBS was mixed with Protein Thermal Shift Dye. The plate was sealed with MicroAmp Optical Adhesive Film (Applied Biosystems, Foster City, CA, USA) and centrifuged at 142× *g* for 1 min. The measurement was performed using a real-time PCR instrument. The instrument was set up according to the manufacturer’s instructions. All the experiments were performed at least in triplicate.

### 2.6. Conversion to IgG and Production of IgG Proteins

The light and heavy chain vectors (pcDNA3.3 and pOptiVEC, respectively) used for Herceptin expression were used as IgG1 backbone vectors. The VL, VH, and CL (light chain variable domain, heavy chain variable domain, and light chain constant domain, respectively) genes were individually amplified using PCR, and then VL and CL were used in an overlap extension PCR. The PCR products (VL-CL, VH) were purified with a QIAquick PCR Purification Kit (QIAGEN, Hilden, Germany) and digested with the following restriction enzymes (New England Biolabs): VL-CL—*Sfi*I and *Eco*RI, VH—*Sfi*I and *Nhe*I. The digestion products were separated on a 1% agarose gel and the single band was purified with a QIAquick Gel Extraction Kit (QIAGEN, Hilden, Germany). The fragments were cloned into *E. coli* DH5α as described in Section 2.3. Individual colonies of the transformed cells were isolated and the sequences of isolated clones were verified.

HEK293F cells were cultured in HEK293F Expression Medium (Thermo Fisher Scientific, Waltham, MA, USA) in a humidified 8% CO_2_ incubator at 37 °C with shaking at 125 rpm. On the day of transfection, the HEK293F cell density was approximately 2.9 × 10^6^ cells/mL. HEK293F transfections were performed using polyethylenimine (PEI) (Polyscience, Niles, IL, USA) according to the manufacturer’s protocol. IgG antibodies were affinity-purified using HiTrap MabSelect SuRe (GE Healthcare, Pittsburgh, PA, USA) columns. Briefly, equilibration was carried out using buffer A (PBS, pH 7.4). The sample was loaded onto the equilibrated column. Following sample loading, the column was washed with buffer A until a stable baseline was established. Following the wash step, the protein was eluted with buffer B (0.1 M glycine, pH 2.7). Following the elution, the IgG was brought to a neutral pH with 1 M Tris base, pH 9.0, and dialyzed into PBS gels (Thermo Fisher Scientific). Purified and buffer-exchanged IgG samples were separated on 4%–12% Bis-Tris gels (Thermo Fisher Scientific) and stained with Sun-Gel Staining Solution (LPS Solution, Deajeon, Korea).

### 2.7. Size-Exclusion Chromatography (SEC) and Intact Mass Analysis 

Separation of the antibodies using size-exclusion chromatography was accomplished using a Waters Alliance 2695 (Waters, Milford, MA, USA) connected to a BioSuite High Resolution SEC Column (7.5 mm × 300 mm, 10 µm particle size, Waters, Milford, MA, USA). The separation was performed using an isocratic elution with PBS, pH 7.0, at a flow rate of 1 mL/min. The effluent was detected with a UV/Vis detector 2469 at 280 nm. 

The intact masses of the antibodies were determined using reverse-phase chromatography separation. The separation for intact mass was accomplished using a Waters ACQUITY I class UPLC system (Waters, Milford, MA, USA) connected to a Thermo MabPac^TM^ RP column (2.1 mm × 50 mm, 4 µm particle size, Thermo Fisher Scientific). The separation was performed with eluent A, consisting of 0.1% formic acid in water, and eluent B, consisting of 0.1% formic acid in 100% acetonitrile, at a flow rate of 0.2 mL/min. The gradient was fixed for 2 min at 20% eluent B and linearly increased for 8 min from 20% to 50% eluent B. The effluent was analyzed with a Waters Synapt G2-Si HDMS system. A quantity of 50 μL (0.2 mg/mL) of antibodies was mixed with 1.5 μL (500,000 units/mL) of PNGaseF (New England BioLabs), then reacted overnight at 37 °C. Native and PNGaseF-treated antibodies were injected with 5 μL of the samples.

### 2.8. Immunofluorescence Microscopy

Cells were cultured in Nunc™ Lab-Tek™ eight-well chamber slides (Thermo Fisher Scientific (Nunc), Rockilde, Denmark). Vero cells were infected with MERS-CoV Erasmus (EMC) strain (kindly provided from Erasmus University, Netherlands). MRC5 cells were infected with human coronaviruses (hCoV) OC-43 strain (ATCC, Manassas, VA, USA) and 229E strain (ATCC). LLC-MK2 cells were infected with hCoV NL63 strain (CN061/14) that was isolated from a Korean patient [36]. After the cytopathogenic effect appeared in the infected cells, the cells were fixed with 70% methanol in PBS (Thermo Fisher Scientific, Carlsbad, CA, USA). Then, human anti-MERS-S2P antibodies (S2A3, S2A6, and S2D5) and a rabbit polyclonal anti-MERS-CoV spike protein antibody, used as a positive control (Sino biological), in TBS (0.1% Triton X-100, 1% BSA in PBS) solution were incubated with the Vero cells infected with MERS-CoV. A mouse anti-229E coronavirus nucleoprotein OC-43 antibody (MERCK, Darmstadt, Germany), a mouse anti-coronavirus antibody, hCoV OC-43 (LifeSpan BioScience, Seattle, WA, USA), and a rabbit polyclonal anti-hCoV-HKU1 spike protein antibody (Sino Biological) were also incubated with MRC5 cells infected with the hCoV 229E or OC-43 strain, and with LLC-MK2 cells infected with the hCoV NL63 strain, with both types of cells used as positive controls. The immunofluorescence assays performed without the positive primary antibodies (described above) were used as negative controls. After incubation with the primary antibodies, the cells were washed with TBS solution. The secondary antibodies, fluorescein isothiocyanate (FITC)-conjugated goat anti-human-IgG (Abcam), FITC-conjugated goat anti-rabbit-IgG (Abcam), and FITC-conjugated goat anti-mouse-IgG (Jackson Immunoresearch, West Grove, PA, USA) were added to the cells and incubated. Cover slips were added on top of the cells after fluorescence mounting medium (Dako, Carpinteria, CA, USA) covered the cells. The cells were observed and images were obtained using a fluorescence microscope (Nikon Eclipse Ti-U, Intensilight C-HGFI, Nikon, Tokyo, Japan). 

### 2.9. Surface Plasmon Resonance (SPR) 

The binding of antibodies to the MERS-S2P was assayed using a Biacore T200 instrument (GE Healthcare, Chicago, IL, USA). An anti-histidine antibody from the His Capture Kit (GE Healthcare) was amine-coupled in the active and reference flow cells of a Sensor Chip CM5 (GE Healthcare) according to the manufacturer’s instructions [37]. Immobilization levels in the range of 14,000 RUs (resonance units) to 15,000 RUs were used, with similar levels in active and reference flow cells. Histidine-tagged MERS-S2P was injected for 30 s using a flow rate of 10 μL/min in the active flow cell only. Capture levels in the range of 70 RUs to 80 RUs were obtained. For kinetic studies, S2A3 (IgG) and S2D5 (IgG) at concentrations ranging from 0.59 nM to 9.38 nM were diluted in HBS-EP+ buffer (GE Healthcare). The experiments were performed at 25 °C with an association time of 3 min and a dissociation time of 5 min at a flow rate of 30 μL/min. After each binding cycle, a regeneration solution (10 mM glycine, pH 1.5) was injected for 1 min at a flow rate of 30 μL/min to remove any non-covalently bound protein. Signal detection was at a rate of 10 signals per second. Binding constants were determined using a 1:1 binding model and the BIA Evaluation software version 1.0 (GE Healthcare).

### 2.10. Neutralization Assay 

Serially four-fold-diluted mAbs and an equal volume of virus (50–100 plaque forming unit (pfu)/well) were incubated at 37 °C for 2 h. The mixtures were added to each well of a 24-well plate seeded with Vero cells (1 × 10^5^ cells/well), and incubated at 37 °C for 1 h. One milliliter of 1.5% carboxymethylcellulose within the medium was overlaid on the cells. After 3 days of incubation, the cells were fixed with 4% paraformaldehyde and stained with crystal violet to visualize the plaques. 90-F1 human monoclonal antibody (KCDC, Cheongju-si, Chungcheongbuk-do, Republic of Korea) was used as a positive control to neutralize MERS-CoV infection. 

### 2.11. Detection of MERS-S2P by ACCEL ELISA™ 

ACCEL ELISA™ plates were coated overnight at 4 °C with 100 μL per well of 3, 5, or 10 μg/mL S2A3 (IgG). After rinsing five times with PBS-T, the wells were blocked with 5% skim milk in PBS for 1 h at RT. The rabbit anti-MERS-CoV antibodies (1:3000, Sino Biological), serially two-fold-diluted MERS-S2P, and HRP-conjugated goat anti-rabbit antibodies (1:6000, Sigma Aldrich) were added and incubated for 30 min at RT. After washing five times with PBS-T, TMB substrate solution was added for 15 min, and the reaction was stopped with 1 N sulfuric acid. Absorbance was measured at 450 nm using a Thermo Scientific MULTISKAN GO Microplate Reader. (Thermo Fisher Scientific). A standard curve was created on a log/log plot using a linear regression algorithm and the limit of detection (LOD) was calculated on a concentration that gave the best linearity (i.e., 5 μg/mL). The plot equation at 5 μg/mL was y = 0.1503x^0.8126^, where the x value was LOD calculated with the y value (y = average absorbance of zero concentration + 3× SD (standard deviation)).

## 3. Results

### 3.1. Selection of Anti-MERS-CoV Fabs 

We have recently constructed a synthetic human Fab phage display library (size ≈1 × 10^10^) based on a human VH3 and Vk1 germline-based framework region (FR) scaffold carrying highly diversified complementarity-determining regions (CDRs) of human antibody heavy and light chains (unpublished data). In order to isolate human antibodies that specifically recognize MERS-S2P, the phage display library was panned against a recombinant MERS-S2P (provided by Dr. Lee from KCDC) immobilized on microtiter plates, followed by evaluation of 94 monoclonal phages from the third round of panning by ELISA (Figure 1A and Appendix A). Twenty-five out of the 94 individual clones had ELISA read-outs at least four times above background levels (skim milk only) and were considered to be potential binders. Fourteen clones were sequenced and determined to be complete and in-frame, but the remaining clones contained mutations such as stop codons and frameshifts (data not shown).

Although the MERS-S2P only shared 40.0%, 34.8%, 44.0%, 33.8%, and 41.5% amino acid identity with other human coronaviruses (hCoV) (SARS-CoV, hCoV-NL63, hCoV-OC43, hCoV-229E, and hCoV-HKU1, respectively) (Appendix A), we investigated whether the Fabs could cross-react with other hCoV-S proteins. To address this, the 14 clones were further analyzed using phage ELISA on recombinant S proteins from SARS-CoV and hCoV-HKU1 (see Materials and Methods). The results showed that the Fabs did not bind other CoV spike proteins, and were monospecific to the MERS-S2P (Figure 1B). By analyzing CDR sequences of the 14 clones, three unique Fab clones (S2A3 (Fab), S2A6 (Fab), and S2D5 (Fab)) were identified. One clone, S2A6 (Fab), was dominantly selected via panning (86% of sequences), while the other two clones (S2A3 (Fab) and S2D5 (Fab)) were present at lower frequencies (7% each) (Figure 1C). In Figure 1B, clones A3 and D5 represent S2A3 (Fab) and S2D5 (Fab), while the other 12 clones represent Fab S2A6.

### 3.2. Production and Characterization of Anti-MERS-CoV Fabs 

To produce and characterize the binders as Fab proteins, the selected sequences were then cloned into an in-house bacterial expression vector (pKFAB). The Fabs were expressed in bacteria and subsequently purified. The resulting Fabs were highly pure with protein yields of 4.4 mg/L, 3.0 mg/L, and 2.5 mg/L for S2A3 (Fab), S2A6 (Fab), and S2D5 (Fab), respectively (Figure 2A and Table 1). 

The two Fab proteins, S2A3 (Fab) and S2D5 (Fab), were observed to be monomeric with no visible high-molecular-weight (HWM) aggregates on size-exclusion chromatography (Figure 2B). However, S2A6 (Fab) were eluted much later on size-exclusion chromatography compared to the two Fabs (S2A3 and S2D5) (Appendix A), although S2A6 (Fab) showed similar migration behavior as the two Fabs on the SDS-PAGE analysis (Figure 2A). This suggests that its abnormal retention time on the size-exclusion chromatography seems not due to the degradation of S2A6 (Fab) protein leading to smaller fragments but rather due to its aggregating, sticky nature. The melting temperatures (*T*_m_, °C) of the Fab proteins were measured using a protein thermal shift (PTS) assay. The results showed that S2A3 (Fab) was the least stable with a *T*_m_ of 61.5 °C, while S2A6 (Fab) and S2D5 (Fab) had higher *T*_m_ values (80.4 °C and 78.2 °C, respectively) (Figure 2C and Table 1). The apparent affinities of the Fabs for MERS-S2P were assessed using ELISA (*EC*_50_, nM). The Fabs had moderate affinities for the S2 subunit (123 nM, 252 nM, and 421 nM for S2A3 (Fab), S2A6 (Fab), and S2D5 (Fab), respectively) (Figure 2D and Table 1).

### 3.3. Production and Characterization of Anti-MERS-CoV IgGs 

In order to produce and characterize the Fab binders in an IgG form, the individual VH and VL sequences from each of the Fab clones were cloned into heavy (IgG1 Fc) and light chain (Ck1) expression vectors, respectively. IgGs were expressed transiently in HEK293 cells and subsequently purified. As shown in Figure 3, the resulting IgGs were highly pure, with protein yields of 17.1 mg/L, 2.7 mg/L, and 12.9 mg/L for S2A3 (IgG), S2A6 (IgG), and S2D5 (IgG), respectively (Figure 3 and Table 1). 

We next assessed whether the MERS-S2P binders were specific to MERS-CoV, which was initially demonstrated by phage ELISA on immobilized spike proteins (Figure 1B). To address this, we used an immunofluorescence assay. As shown in Figure 4A, S2A3 (IgG) and S2D5 (IgG) were specific for MERS-CoV: the two IgG antibodies bound only MERS-CoV but showed no indication of binding to other coronavirus strains or cell lines tested (Figure 4A). However, while S2A6 (IgG) did not bind to the other coronavirus strains and their cell lines, it appeared to bind non-specifically to the Vero cells that were used for MERS-CoV infection. 

To determine whether S2A3 (IgG) and S2D5 (IgG) were free of aggregates, they were analyzed using size-exclusion chromatography (Figure 4B). S2D5 (IgG) did not form any aggregates, while S2A3 (IgG) contained ≈5.7% high molecular weight aggregates (Figure 4B). Next, to demonstrate whether the IgG antibodies were produced in an intact form, S2A3 (IgG) and S2D5 (IgG) were analyzed using liquid chromatography–mass spectrometry (LC-MS) with and without PNGase F treatment. The results confirmed that all of the observed masses were within 15 Da of the theoretical masses of the intact IgGs, indicating that the IgG antibodies were produced with very few modifications (Appendix A). The apparent binding affinities (*K*_D_, M) of S2A3 (IgG) and S2D5 (IgG) for MERS-S2P were determined using the Biacore T200. As shown in Figure 4, S2A3 (IgG) and S2D5 (IgG) showed 170 pM and 1.19 nM apparent affinities for MERS-S2P, respectively (Figure 4C and Table 1), in this experiment, indicating that a significant interaction occurred.

### 3.4. Detection of MERS-S2P Using ACCEL ELISA^TM^

Since the two IgG antibodies (S2A3 (IgG) and S2D5 (IgG)) failed to neutralize MERS-CoV, i.e., the two mAbs could not prevent MERS-CoV infection, while the viral infection was blocked with a positive control mAb (90-F1) (Appendix A), we investigated their application as reagents for the detection of anti-MERS-CoV IgG in serum samples. To address this, we performed a “proof of concept” study based on a sandwich immunoassay using the ACCEL ELISA™ system, a rapid and sensitive ELISA system developed by Plexense Inc. (Yongin-si, Gyeonggi-do, Republic of Korea). S2A3 (IgG) and a rabbit polyclonal anti-MERS-CoV IgG were used as antigen capture and detection reagents, respectively (Figure 5A). First, three different concentrations of S2A3 (IgG) (3 μg/mL, 5 μg/mL, and 10 μg/mL) were employed to determine which one provided the best linearity or “goodness of fit” in the sandwich ACCEL ELISA™. As shown in Figure 5B, the best linearity (R^2^ = 0.9962) was obtained when S2A3 (IgG) was immobilized at a concentration of 5 μg/mL (Figure 5B). Under this condition, the limit of detection (LOD) of MERS-S2P detectable in the ELISA was calculated to be 0.33 ng/mL (≈5 fmol/mL) (Figure 5B), suggesting that S2A3 (IgG) can be applied to the ACCEL ELISA™ as a capture reagent to diagnose MERS-CoV infection through the detection of MERS-S2P.

## 4. Discussion

We report the selection of human mAbs specific to the MERS-CoV S2 subunit (MERS-S2P) using a human synthetic Fab phage display library. Phage display is a powerful tool that has been used for both the discovery of, and therapeutic applications against, infectious diseases [14,18,39,40]. In particular, phage display has been demonstrated to be highly effective for the selection of human antibodies against MERS-CoV, mostly in naïve human antibody phage display formats generated from human peripheral blood mononuclear cells (PBMCs) [18]. Although, in general, human mAbs targeting the RBD of the S1 subunit have higher neutralizing potencies than those targeting other regions of the S protein, such as HR1 on the S2 subunit, it is still necessary to combine human mAbs that recognize different neutralizing epitopes due to the emergence of viruses carrying RBD mutations [19,20]. Mouse G4 mAb is the only mAb demonstrated to bind MERS-S2P with an affinity of 8.9 nM and to neutralize infection of pseudo-MERS-CoV bearing S protein [21]. Recently, it was revealed that the G4 mAb recognizes the connector domain present between the HR1 and HR2 domains of the S2 subunit. This region is variable in sequence and length among betacoronaviruses [13]. Although the two anti-MERS-S2P mAbs (S2A3 and S2D5) selected from our phage display library failed to neutralize infection by MERS-CoV (data not shown), these represent the first human antibodies targeting MERS-S2P selected from a human synthetic phage display library. Moreover, they had desirable biophysical properties in terms of affinity, thermal stability, and non-aggregation, which are essential for the development of therapeutics and diagnostics. Furthermore, the reformatting of the Fabs to IgGs highly increased their apparent affinity (*K*_Dapp_) to MERS-S2P, seemingly due to the avidity effect. Thus, these antibodies represent valuable additional S2 subunit-targeting mAbs and it should be interesting to reveal where they bind on the S2 subunit through a structural study, which is currently ongoing. 

Based on the sub-nM apparent affinity of S2A3 (IgG) and its specificity to MERS-CoV, we tried to develop an anti-MERS-S2P-based antibody capture ELISA for the diagnosis of MERS-CoV infection. Once again, this is the first ELISA-based assay employing an anti-MERS-S2P mAb. The ELISA was tested using ACCEL ELISA™ (Plexense Inc.), which is a rapid and sensitive ELISA system in which each well of the ELISA plate is stacked with layers such that the surface area of the well is increased. This results in higher ELISA signals because more protein is immobilized on the surface, and also enables faster development (<20 min) with a smaller amount of reagents (≈15 μL) due to the effects of stacking. Indeed, our proof of concept study with S2A3 (IgG), MERS-S2P and rabbit polyclonal anti-MERS-CoV IgGs demonstrated a detection limit of 0.33 ng/mL (≈5 fmol/mL). Previously, we have tested S2D5 (Fab) to determine the feasibility of using this mAb on a paper-based analytical device (PAD) designed to capture MERS-S2P. We observed that the mAb worked effectively on the PAD and the detection limit for MERS-S2P was approximately 10.4 nM (≈160 fmol/mL) [41]. Due to the lack of MERS-S2P-based detection assays that are comparable with ours, it is not possible to evaluate our results directly. However, it is still possible to compare our results indirectly with those obtained from detection assays performed using nucleocapsid protein (NP) through an antigen capture ELISA-based method or an immunochromatographic test (ICT) [28,30]. Similar to SARS-CoV, the NP protein is preferable as a target to the S protein in MERS, simply due to the presence of a higher copy number of the NP protein than the S protein [30]. According to the previous NP-based studies, a recombinant NP capture ELISA-based method showed detection limits of 0.625 ng/mL (≈13.6 fmol/mL) and of 5 ng/mL (≈108.7 fmol/mL) using an ICT [30], while another capture ELISA yielded a detection limit of 1 ng/mL (≈21.7 fmol/mL) [28]. This highlights the fact that our MERS-S2P-based ACCEL ELISA™ system performed better than previously developed methods, even though the target protein is different. Studies are currently underway to demonstrate whether the MERS-S2P-based ACCEL ELISA™ can feasibly detect virions in anti-MERS-CoV serum, and if so, to identify the detection limit in TCID_50_/mL (TCID_50_, 50% tissue culture infective dose). This should provide a direct comparison with the NP capture ELISA-based assays. It would also be interesting to perform an ACCEL ELISA™ with anti-NP mAbs to compare their performance in a more quantitative way.

The MERS-S2P-based ACCEL ELISA™ assay also could be applied to detect anti-MERS-S2P IgGs from anti-viral serum samples of MERS-CoV-infected people. A drawback of this strategy is that, in general, antibodies against the virus appear within 10 days of the onset of illness; therefore, antibodies might fall behind the detection of viral genomic material using molecular methods such as RT-PCR [42]. Thus, the MERS-S2P-based ACCEL ELISA™ assay should be performed in combination with molecular methods, thereby detecting the viral genome as the front line of the diagnosis, such that viral infection can be diagnosed by molecular methods using lower respiratory tract samples where the viral load is high. Later, or in case these methods fail to detect viral genomes in the respiratory samples, the MERS-S2P-based ELISA assay, alone or possibly in combination with an NP-based antigen capture assay, can be applied to confirm and diagnose viral infection. Another point to address in this strategy is that since our anti-MERS-S2P mAbs are human antibodies, they should be dehumanized to avoid false positives due to their binding to the secondary anti-human antibodies before they are applied in the assay. Given the high titers of antibodies against MERS-CoV in over 95% of adult dromedaries, the immediate reservoir for the MERS-CoV transmission to humans, namely the Middle East [43,44,45], the present MERS-S2P-based antibody capture ELISA may be potentially useful for detecting MERS-CoV in dromedary animal samples. It may also be useful for screening other animal samples for MERS-CoV as long as secondary antibodies detecting the animal-derived anti-MERS-S2P antibodies are available. 

In conclusion, we have selected high-affinity human anti-MERS-S2P mAbs from a human synthetic phage display library. We characterized the resulting Fabs and IgGs to observe their high affinity to MERS-S2P, high specificity to MERS-CoV, and desirable biophysical properties. These properties warrant further development. We tested S2A3 (IgG), a MERS-S2P mAb, in ACCEL ELISA™ to develop the first MERS-S2P-based antibody capture assay. The detection limit of this assay surpassed that of previously developed NP-based antigen capture assays. Further refinement of the test should warrant the development of a rapid, sensitive detection kit for a point of care test. 

## 5. Patent 

The full-length antibody sequences of the human anti-MERS-S2P Fabs are patent-pending subject matter in Korea, patent number 10-1969696 (registration date April 10, 2019).

## Figures and Tables

**Figure 1 antibodies-08-00042-f001:**
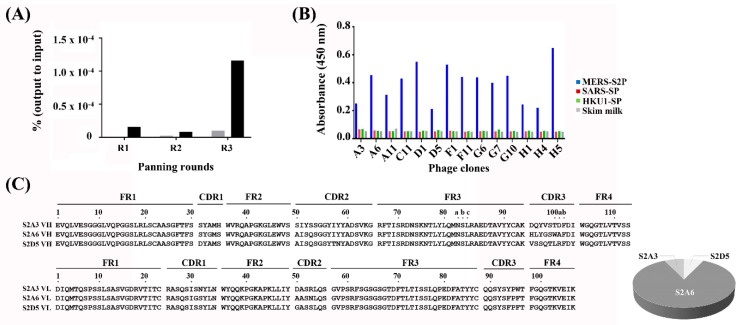
Output of the panning of the phage-displayed synthetic Fab library on MERS-S2P. (**A**) Monitoring of phage titers over three rounds (R1–R3) of panning. Black and gray bars indicate a ratio of phage output to input titers presented as a percentage (%) from panning on MERS-S2P-immobilized and -non-immobilized surfaces, respectively. The ratio of output to input (%) = (phage output titer ÷ phage input titer) × 100. (**B**) Phage ELISAs performed on MERS-S2P-, SARS-CoV spike protein-, a CoV spike protein-immobilized surfaces (blue, red, and green, respectively). (**C**) Amino acid sequences of three unique clones identified from panning (left) and their relative frequencies (%) (right). The sequences were aligned using the Kabat numbering system [38]. ELISA, enzyme-linked immunosorbent assay; MERS-S2P, Middle East respiratory syndrome-CoV S2 subunit protein; SARS-SP, severe acute respiratory syndrome-CoV S protein; HKU1-SP, hCoV HKU1 S protein; CoV, coronavirus; CDR, complementarity-determining region; FR, framework region.

**Figure 2 antibodies-08-00042-f002:**
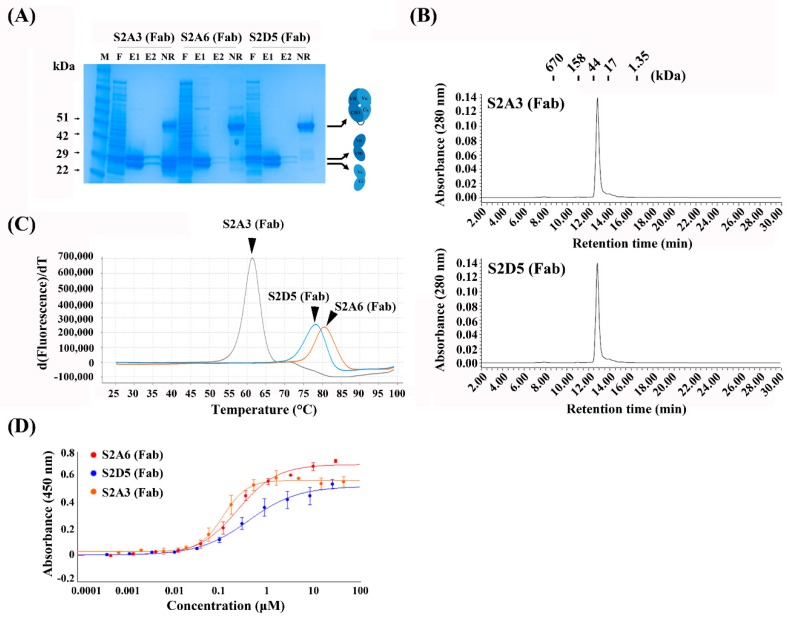
Production and characterization of anti-MERS-S2P Fabs. (**A**) SDS-PAGE analysis of anti-MERS-S2P Fabs purified from periplasmic extracts of *E. coli* transformed with the indicated expression vectors. (**B**) Size-exclusion chromatography analysis of S2A3 (Fab) and S2D5 (Fab). The positions of the molecular mass markers on the retention time x-axis are shown above the peak. (**C**) Protein thermal shift assay of anti-MERS-S2P Fabs to determine their thermal stability (*T*m, °C). (**D**) Soluble ELISA of serially-diluted anti-MERS-S2P Fabs on MERS-S2P-immobilized surfaces to measure their apparent affinities (*EC*50, μM). Fab, antigen-binding fragment; SDS-PAGE, sodium dodecyl sulfate-polyacrylamide gel electrophoresis; M, molecular mass marker; F, flow-through; E1 and E2; the first and second elutes, respectively; DTT, 1,4-Dithiothreitol; NR, non-reducing (no DTT added).

**Figure 3 antibodies-08-00042-f003:**
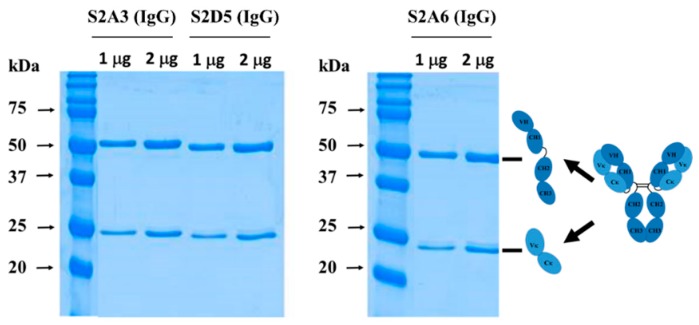
Production of anti-MERS-S2P IgGs. SDS-PAGE analysis under reducing conditions (with DTT) of anti-MERS-S2P IgGs purified from culture media of HEK293F cells transiently transfected with heavy- and light-chain expression vectors. IgG, immunoglobulin G.

**Figure 4 antibodies-08-00042-f004:**
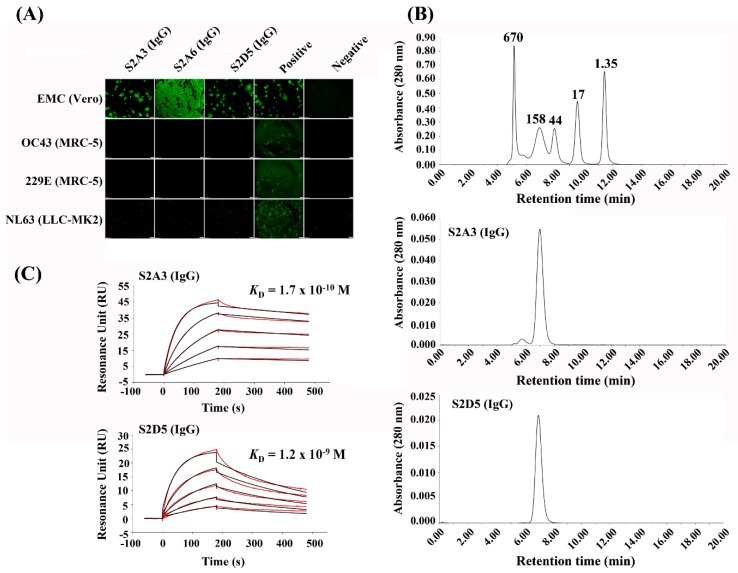
Characterization of anti-MERS-S2P IgGs. (**A**) Immunofluorescence assay of anti-MERS-S2P IgGs on cell lines infected with various human coronaviruses (hCoVs) to determine their MERS-CoV specificity. Scale bar, 200 μm. (**B**) Size-exclusion chromatography analysis of S2A3 (IgG) and S2D5 (IgG). The molecular weights (kDa) of the molecular mass markers are shown above the corresponding peaks at the top chromatogram. (**C**) Surface plasmon resonance (SPR) analysis of S2A3 (IgG) and S2D5 (IgG) on a MERS-S2P-immobilized sensor chip to determine their apparent binding affinities. The fitted-lines are marked by red.

**Figure 5 antibodies-08-00042-f005:**
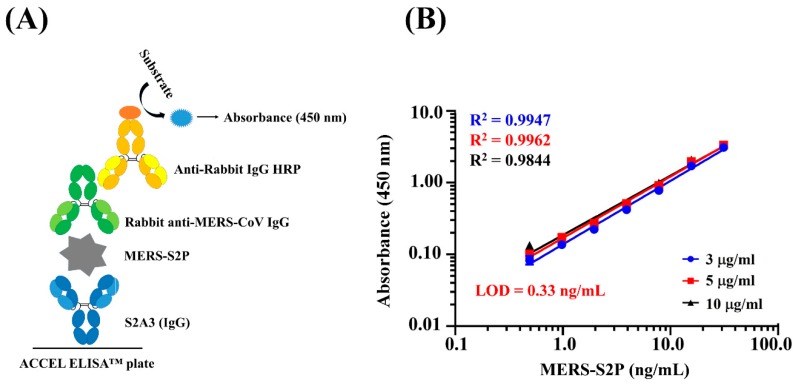
Detection of MERS-S2P using S2A3 (IgG) on ACCEL ELISA™ plates. (**A**) Schematic depicting the sandwich ELISA format to detect MERS-S2P using S2A3 (IgG) (capture antibody) and rabbit anti-MERS-CoV IgG (detection antibody) on ACCEL ELISA™ plates. (**B**) ELISA detection of MERS-S2P on a capture antibody (S2A3 (IgG)) immobilized using three different concentrations (3 µg/mL, 5 µg/mL, and 10 µg/mL) on ACCEL ELISA™ plates. The goodness of fit is indicated by the R^2^ value. LOD, limit of detection.

**Table 1 antibodies-08-00042-t001:** Physicochemical properties of human anti-MERS-S2P monoclonal antibodies (mAbs).

Clones	Yield (mg/L)	*T*_m_ (°C)	Monomericity	*EC_50_* (nM)	*k*a (1/MS)	*k*d (1/S)	*K*_D_ (M)
**S2A3 (Fab)**	4.4	61.5	Mon.	123	n.d.	n.d.	n.d.
**S2A6 (Fab)**	3.0	80.4	Agg.	252	n.d.	n.d.	n.d.
**S2D5 (Fab)**	2.5	78.2	Mon.	421	n.d.	n.d.	n.d.
**S2A3 (IgG)**	17.1	n.d.	Mon. with 5.7% agg.	n.d.	2.5 × 10^6^	4.4 × 10^−4^	1.7 × 10^−10^
**S2A6 (IgG)**	2.7	n.d.	n.d.	n.d.	n.d.	n.d.	n.d.
**S2D5 (IgG)**	12.9	n.d.	Mon.	n.d.	2.2 × 10^6^	2.6 × 10^−3^	1.2 × 10^−9^

SEC, size-exclusion chromatography; IgG, immunoglobulin G; n.d, not determined; *T*_m_, melting temperature; *EC*_50_, half maximal effective concentration; *k*a, association rate constant; *k*d, dissociation rate constant; *K*_D_, equilibrium dissociation constant. Mon., Monomer; Agg., Aggregate.

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
