# Peer review of "Selection and Characterization of Monoclonal Antibodies Targeting Middle East Respiratory Syndrome Coronavirus through a Human Synthetic Fab Phage Display Library Panning"

_2073-4468, 2019, doi:10.3390/antib8030042_

Round 1

Reviewer 1 Report

The authors describe the isolation of three antibodies specific for the S2 subunit of MERS-CoV spike protein, using biopanning of a naive synthetic human Fab phage display library. The work is significant since diagnotics and therapeutics of MERS-CoV infection are still limited. Diagnostics could be improved using monoclonal antibodies for faster point-of-care assays, rather than relying on RT-PCR and viral RNA sequencing. The authors claim novelty in that they are the first to isolate antibodies against the S2 subunit using phage display, and that the S2 domain is less prone to viral mutation than the S1 domain. Previous antibodies to the S2 subunit have been isolated using mouse immunisation. Unfortunately the antibodies are not neutralising so they are not useful for therapeutic use, but they may make good diagnostic reagents.

Specific points:

Major issues:

Section 2.1:  Please provide more information about the antigen used for biopanning. Currently it just states that the S2 subunit of MERS-CoV spike protein was provided by Dr Lee. Is it recombinant or isolated from viral stocks? Does it have any tags that need to be addressed during biopanning (eg a His tag could potentially yield His-tag binders).

Section 2.3: Please give a description of the Fab expression vectors, to indicate that a Strep tag is introduced in the Fab fragment. This is important to understand the purification strategy.

Section 2.7: Why were the VL and CL genes amplified separately, and then spliced back together using SOE PCR? If the library is a Fab library, aren't the VL and CL genes already together, and could be amplified as an intact unit?

Section 3.4:  I am confused by the design of the ELISA. For an antibody detection assay, couldn't the MERS-S2P protein be directly immobilised onto the ELISA plate, rather than being captured by the S2A3 antibody? Capture via the antibody could potentially obsure immunogenic sites, competing with binding of serum antibodies. Binding of serum antibodies could also cause dissociation of the MERS-S2P-S2A3 interaction.  Potentially, the S2A3 antibody could be useful in an antigen detection assay, rather than an antibody detection assay, to detect virions in serum?

Figure 5B:  Limit of detection is now shown on the graph (all points are still linear), yet the LOD is given in the text.  How was the LOD calculated?

Minor Issues:

Abstract:  Undefined definition (EMC)

Introduction, first sentence: The statement 'a novel human coronavirus (hCoV)' is not necessary - it is redundant to other statements in the sentence.

Introduction, last sentence of first paragraph:  'The overall risk in areas outside Saudi Arabia, the country in which most of the cases have been reported, remains significant, and thus novel tools and reagents for therapy and diagnosis are urgently needed.'  Please rephrase this sentence - it sounds like cases within Saudi Arabia are not significant to warrant new therapeutics and diagnostics.

Section 2.1: 'mid-log phage E.coli TG1 cells'.  Should be 'mid-log phase E.coli TG1 cells.

Section 2.1: Centrifuge speeds should be reported in 'g' rather than rpm

Section 2.6: Perhaps move this section to after the conversion to IgG, since the intact mass analysis was only done on the full IgGs

Figure 1B: Y-axis label should be 450nm not 280nm 

Figure 2B and S2: Please give a size indication on the Size Exclusion chromatograms. Why does S2A6 Fab have a longer retention time (smaller size) than S2A3 and S2D5?

Discussion, second paragragh: 'Previously, we have tested S2D5 (Fab), another anti-MERS-S2P mAb, to determine the feasibility......(ref 41)'.  Is this the same S2D5 described in this paper? Ref 41 calls it D5, but obtained it from the same group (New Drug Development Center).  If it's the same antibody, delete the statement 'another anti-MERS-S2P mAb', as this is confusing.

Reviewer 2 Report

The topic of the manuscript is important. The manuscript is well-written and interesting. There are several relatively minor things that need to be corrected prior to publication.

Abstract

Correct “EC50 = 123 nM-421 nM” to read “EC50 = 123-421 nM”

Introduction

Say either “spike protein” or “S protein”, not “S spike protein” (see 2nd paragraph).

The Authors need to mention such disadvantage of monoclonal antibody therapies as their high cost.

Materials and Methods

Monoclonal phage ELISA

How many individual phage clones were tested for binding in ELISA?

Results

What was the total number of individual phage clones tested in ELISA?

Font size in Figure 1 is too small.

Figure 1a: Present data as ratio of output to input phage expressed in %.

Font size in Figure 2 is too small.

Need clarification as to Figure 4A. What were the positive and negative controls?  The Authors indicate in the text non-specific staining of Vero cells. How did this influence their conclusion about specificity of MERS-S2P binders to MERS-CoV?

Font size in Figure 4 is too small.

Section 3.4

Include results that show that S2A3 and S2D5 antibodies failed to neutralize MERS-CoV. What method was used?

No reference to Figure 5A in the text.

Discussion

Do not reference Figures in the Discussion section.
